# On Scalable Testing of Samplers [*][†]

**Yash Pote**   (r)   **Kuldeep S. Meel**
School of Computing, National University of Singapore

## Abstract

In this paper we study the problem of testing of constrained samplers over high-dimensional distributions with $(\varepsilon, \eta, \delta)$ guarantees. Samplers are increasingly used in a wide range of safety-critical ML applications, and hence the testing problem has gained importance. For $n$-dimensional distributions, the existing state-of-the-art algorithm, Barbarik2, has a worst case query complexity of exponential in $n$ and hence is not ideal for use in practice. Our primary contribution is an exponentially faster algorithm that has a query complexity linear in $n$ and hence can easily scale to larger instances. We demonstrate our claim by implementing our algorithm and then comparing it against Barbarik2. Our experiments on the samplers wUnigen3 and wSTS, find that Barbarik3 requires $10\times$ fewer samples for wUnigen3 and $450\times$ fewer samples for wSTS as compared to Barbarik2.

## 1 Introduction

The constrained sampling problem is to draw samples from high-dimensional distributions over constrained spaces. A constrained sampler $\mathcal{Q}(\varphi, \mathtt{w})$ takes in a set of constraints $\varphi : \{0,1\}^n \to \{0,1\}$ and a weight function $\mathtt{w} : \{0,1\}^n \to \mathbb{R}_{>0}$, and returns a sample $\sigma \in \varphi^{-1}(1)$ with probability proportional to $\mathtt{w}(\sigma)$. Constrained sampling is a core primitive of many statistical inference methods used in ML, such as Sequential Monte Carlo[29], Markov Chain Monte Carlo(MCMC)[3, 9], Simulated Annealing [4], and Variational Inference [24]. Sampling from real-world distributions is often computationally intractable, and hence, in practice, samplers are heuristical and lack theoretical guarantees. For such samplers, it is an important problem to determine whether the sampled distribution is close to the desired distribution, and this problem is known as *testing of samplers*. The problem was formalised in [14, 27] as follows: Given access to a target distribution $\mathcal{P}$, a sampler $\mathcal{Q}(\varphi, \mathtt{w})$, and three parameters $(\varepsilon, \eta, \delta)$, with probability at least $1 - \delta$, return (1) Accept if $d_\infty(\mathcal{P}, \mathcal{Q}(\varphi, \mathtt{w})) < \varepsilon$, or (2) Reject if $d_{TV}(\mathcal{P}, \mathcal{Q}(\varphi, \mathtt{w})) > \eta$. Here $d_{TV}$ is the total variation distance, $d_\infty$ the multiplicative distance, and $\varepsilon, \eta$, and $\delta$ are parameters for closeness, farness and confidence respectively. Access to distribution $\mathcal{P}$ is via the DUAL oracle, and access to $\mathcal{Q}$ is via the PCOND and SAMP oracles (defined in Section 2.1)

There is substantial interest in the testing problem due to the increasing use of ML systems in real-world applications where safety is essential, such as medicine [2], transportation [8, 25], and warfare [28]. For the ML systems that incorporate samplers, the typical testing approach has been to show the convergence of the sampler with the target distribution via empirical tests that rely on heuristics and do not provide any guarantees [19, 22, 31, 34]. In a recent work [27], a novel framework, called Barbarik2, was proposed that could test a given sampler while providing $(\varepsilon, \eta, \delta)$ guarantees, using $\tilde{O}\left(\frac{tilt(\mathcal{P})^2}{\eta(\eta-3\varepsilon)^3}\right)$ queries, where $tilt(\mathcal{P}) := \max\limits_{\sigma_1, \sigma_2 \in \{0,1\}^n} \frac{\mathcal{P}(\sigma_1)}{\mathcal{P}(\sigma_2)}$ for $\mathcal{P}(\sigma_2) > 0$. Since the $tilt(\mathcal{P})$

---

[*]The accompanying tool, available open source, can be found at https://github.com/meelgroup/barbarik

[†]The authors decided to forgo the old convention of alphabetical ordering of authors in favor of a randomized ordering, denoted by (r). The publicly verifiable record of the randomization is available at https://www.aeaweb.org/journals/policies/random-author-order/search with confirmation code: Lrr1ecP-xv14. For citations, the authors request that the citation guidelines by AEA for random author ordering be followed.

36th Conference on Neural Information Processing Systems (NeurIPS 2022).

can take arbitrary values, we observe that the query complexity can be prohibitively large[3]. On the other hand, the best known lower bound for the problem, derived from [30], is $\tilde{\Omega}\left(\frac{\sqrt{n/\log(n)}}{\eta^2}\right)$. In this work, we take a step towards bridging this gap with our algorithm, Barbarik3, that has a query complexity of $\tilde{O}\left(\frac{\sqrt{n}\log n}{(\eta-11.6\varepsilon)\eta^3} + \frac{n}{\eta^2}\right)$, representing an exponential improvement over the state of the art.

To be of any real value, testing tools must be able to scale to larger instances. In the case of constrained samplers, the only existing testing tool, Barbarik2, is not scalable owing to its query complexity. The lack of scalability is illustrated by the following fact: product distributions are the simplest possible constrained distributions, and given a union of two $n$-dimensional product distributions, Barbarik2 requires more than $10^8$ queries for $n > 30$. On the other hand, the query complexity of Barbarik3 scales linearly with $n$, the number of dimensions, thus making it more appropriate for practical use.

We implement Barbarik3 and compare it against Barbarik2 to determine their relative performance. In our experiments, we consider two sets of problems, (1) constrained sampling benchmarks, (2) scalable benchmarks and two constrained samplers wSTS and wUnigen3. We found that to complete the test Barbarik3 required at least $450\times$ fewer samples from wSTS and $10\times$ fewer samples from wUnigen3 as compared to Barbarik2. Moreover, Barbarik3 terminates with a result on at least $3\times$ more benchmarks than Barbarik2 in each experiment.

Our contributions can be summarized as follows:

1. For the problem of testing of samplers, we provide an exponential improvement in query complexity over the current state of the art test Barbarik2. Our test, Barbarik3, makes a total of $\tilde{O}\left(\frac{\sqrt{n}\log n}{(\eta-11.6\varepsilon)\eta^3} + \frac{n}{\eta^2}\right)$ queries, where $\tilde{O}$ hides polylog factors of $\varepsilon, \eta$ and $\delta$.

2. We present an extensive empirical evaluation of Barbarik3 and contrast it with Barbarik2. The results indicate that Barbarik3 requires far fewer samples and terminates on more benchmarks when compared to Barbarik2.

We define the notation and discuss related work in Section 2. We then present the main contribution of the paper, the test Barbarik3, and its proof of correctness in Section 3. We present our experimental findings in Section 4 and then we conclude the paper and discuss some open problems in Section 5. Due to space constraints, we defer some proofs and the full experimental results to the supplementary Section A and B respectively.

## 2 Notation and preliminaries

**Probability distributions** In this paper we deal with samplers that sample from discrete probability distributions over high-dimensional spaces. We consider the sample space to be the $n$-dimensional Boolean hypercube $\{0,1\}^n$. A constrained sampler $\mathcal{Q}$ takes in a set of constraints $\varphi : \{0,1\}^n \to \{0,1\}$ and a weight function $\mathtt{w} : \{0,1\}^n \to \mathbb{R}_{>0}$, and samples from the distribution $\mathcal{Q}(\varphi, \mathtt{w})$ defined as $\mathcal{Q}(\varphi, \mathtt{w})(\sigma) = \begin{cases} \mathtt{w}(\sigma)/\mathtt{w}(\varphi) & \sigma \in \varphi^{-1}(1) \\ 0 & \sigma \in \varphi^{-1}(0) \end{cases}$, where $\mathtt{w}(\varphi) = \sum_{\sigma \in \varphi^{-1}(1)} \mathtt{w}(\sigma)$. To improve readability, we use $\mathcal{Q}$ to refer to the distribution $\mathcal{Q}(\varphi, \mathtt{w})$. For an element $i$, $\mathcal{D}(i)$ denotes it's probability in distribution $\mathcal{D}$ and $i \sim \mathcal{D}$ represents that $i$ is sampled from $\mathcal{D}$. For any non-empty set $S \subseteq \{0,1\}^n$, $\mathcal{D}_S$ is the distribution $\mathcal{D}$ conditioned on set $S$, and $\mathcal{D}(S)$ is the probability of $S$ in $\mathcal{D}$ i.e., $\mathcal{D}(S) = \sum_{i \in S} \mathcal{D}(i)$.

The total variation (TV) distance of two probability distributions $\mathcal{D}_1$ and $\mathcal{D}_2$ is defined as: $d_{TV}(\mathcal{D}_1, \mathcal{D}_2) = \frac{1}{2}\sum_{i \in \{0,1\}^n} |\mathcal{D}_1(i) - \mathcal{D}_2(i)|$. For $S \subseteq \{0,1\}^n$, we define $d_{TV(S)}(\mathcal{D}_1, \mathcal{D}_2) = \frac{1}{2}\sum_{i \in S} |\mathcal{D}_1(i) - \mathcal{D}_2(i)|$. The multiplicative distance of $\mathcal{D}_2$ from $\mathcal{D}_1$ is defined as: $d_\infty(\mathcal{D}_1, \mathcal{D}_2) = \max_{i \in \{0,1\}^n} |\mathcal{D}_2(i)/\mathcal{D}_1(i) - 1|$. The two notions of distance obey the identity: $2d_{TV}(\mathcal{D}_1, \mathcal{D}_2) \leq d_\infty(\mathcal{D}_1, \mathcal{D}_2)$.

In the rest of the paper, $\mathbb{E}[v]$ represents the expectation of random variable $v$ and $[k]$ represents the set $\{1, 2 \ldots, k\}$.

---

[3]A simple modification reveals that in terms of $n, \eta, \varepsilon$, the bound is $\tilde{O}\left(\frac{4^n}{\eta(\eta-3\varepsilon)^3}\right)$

**Tools used in the analysis**

**Proposition 1** (Hoeffding). *For independent 0-1 random variables $X_i$, $X = \sum_{i=1}^{k} X_i$, and $t \geq 0$,* $\Pr(X - \mathbb{E}X > t) \leq \exp\left(-2t^2 k\right)$ *and* $\Pr(\mathbb{E}X - X > t) \leq \exp\left(-2t^2 k\right)$

**Proposition 2** (Chebyshev). *Given bounded r.v. $X$, we have* $\Pr(|X - \mathbb{E}[X]| < \mathbb{E}[X]) > \frac{\mathbb{E}[X]^2}{\mathbb{E}[X^2]}$

**Proposition 3.** *Given distributions $\mathcal{D}_1$ and $\mathcal{D}_2$ supported on $\{0,1\}^n$, and a set $S \subseteq \{0,1\}^n$,*

$$\sum_{i \in S} \mathcal{D}_1(i)\mathcal{D}_2(i) > \frac{(\mathcal{D}_1(S) + \mathcal{D}_2(S) - 2d_{TV(S)}(\mathcal{D}_1, \mathcal{D}_2))^2}{4|S|}$$

The proof can be found in the Appendix A.1 □

If we are given samples $\{s_1, s_2, \ldots, s_n\}$ from a distribution $\mathcal{D}$ over $[k]$, then the empirical distribution $\widehat{\mathcal{D}}$ is defined to be $\widehat{\mathcal{D}}(i) = \frac{1}{n} \sum_{j=1}^{k} \mathbb{1}_{\{s_j = i\}}$.

**Proposition 4** (See [11] for a simple proof). *Suppose $\mathcal{D}$ is a distribution over $[k]$, and $\widehat{\mathcal{D}}$ is constructed using* $\max\left(\frac{k}{\eta^2}, \frac{2\ln(2/\delta)}{\eta^2}\right)$ *samples from $\mathcal{D}$. Then $d_{TV}(\mathcal{D}, \widehat{\mathcal{D}}) \leq \eta$ with probability at least $1 - \delta$.*

## 2.1 Testing with the help of oracles

In distribution testing, we are given samples from an unknown distribution $\mathcal{P}$ over a large support $\{0,1\}^n$, and the task is to test whether $\mathcal{P}$ satisfies some property of interest. One of the important properties we care about is whether $\mathcal{P}$ is close to another distribution $\mathcal{Q}$, and this subfield of testing is known as *closeness testing*. It was shown by Valiant and Batu et al. that the ability to draw samples from $\mathcal{P}$ and $\mathcal{Q}$ is not powerful enough, as at least $\Omega(2^{2n/3})$ samples are required to provide any sort of probabilistic guarantee for closeness testing. Since $n$ is usually large, it was desirable to find tests that could solve the closeness testing problem using polynomially many samples in $n$.

Motivated by the above requirement, Canonne et al. and Chakraborty et al. introduced the *conditional sampling oracle* (COND), that is a more powerful way to access distributions. A COND oracle for distribution $\mathcal{D}$ over $\{0,1\}^n$ takes as input a set $S \subseteq \{0,1\}^n$ with $\mathcal{D}(S) > 0$, and returns a sample $i \in S$ with probability $\mathcal{D}(i)/\mathcal{D}(S)$. It has been shown that the use of the COND oracle, and its variants, drastically reduces the sample complexity of many tasks in distribution testing [1, 21, 12, 15, 6, 26, 7, 17, 13, 30] (see [10] for an extensive survey). In this paper, we consider the pair-conditioning (PCOND) oracle, which is a special case of the COND oracle with the restriction that $|S| = 2$ i.e., the size of the conditioning set has to be two. To engineer practical PCOND oracle access into constrained samplers, we use the chain formula construction introduced in [14].

With the same goal of designing tests with polynomial sample complexity, a different kind of oracle, known as the DUAL oracle, was proposed by Canonne et al.. The DUAL oracle allows one to sample from a given distribution and also query the distribution for the probability of arbitrary elements of the support. Tractable DUAL oracle access is supported by a number of distribution representations, such as the fragments of probabilistic circuits (PC) that support the EVI query [18]. In our experimental evaluation, we use distributions from one such fragment: weighted d-DNNFs. Weighted d-DNNFs are a class of arithmetic circuits with properties that enable DUAL oracle access in time linear in the size of the circuit [16, 23].

## 3 Barbarik3: an algorithm for testing samplers

We start by providing a brief overview of our testing algorithm before providing the full analysis.

### 3.1 Algorithm outline

The pseudocode of Barbarik3 is given in Algorithm 1. We adapt the definition of bucketing of distributions from [30] for use in our analysis.

**Definition 1.** *For a given $k \in \mathbb{N}_{>0}$, the bucketing of $\{0,1\}^n$ with respect to $\mathcal{P}$ is defined as follows: For $1 \leq i \leq k$, let $S_i = \{b : 2^{-i} < \mathcal{P}(b) \leq 2^{-i+1}\}$ and let $S_0 = \{0,1\}^n \setminus \bigcup_{i \in [k]} S_i$. Given*

---

**Algorithm 1** Barbarik3$(\mathcal{P}, \mathcal{Q}, \eta, \varepsilon, \delta)$

---

1: $k \leftarrow n + \lceil \log_2(100/\eta) \rceil$
2: **for** $i = 1$ to $k$ **do**
3: $\quad S_i = \{b : 2^{-i} < \mathcal{P}(b) \leq 2^{-i+1}\}$
4: $S_0 = \{0,1\}^n \setminus \bigcup_{i \in [k]} S_i$
5: $B_{\mathcal{P}}$ is the distribution over $[k] \cup \{0\}$ where we sample $i \sim B_{\mathcal{P}}$ if we sample $j \sim \mathcal{P}$ and $j \in S_i$
6: $B_{\mathcal{Q}}$ is the distribution over $[k] \cup \{0\}$ where we sample $i \sim B_{\mathcal{Q}}$ if we sample $j \sim \mathcal{Q}$ and $j \in S_i$
7: $\theta \leftarrow \eta/20$
8: $\widehat{d} \leftarrow$ OutBucket$(B_{\mathcal{P}}, B_{\mathcal{Q}}, k, \theta, \delta/2)$
9: **if** $\widehat{d} > \varepsilon/2 + \theta$ **then**
10: $\quad$ **Return** Reject
11: $\varepsilon_2 \leftarrow \widehat{d} + \theta$
12: **Return** InBucket$(\mathcal{P}, \mathcal{Q}, k, \varepsilon, \varepsilon_2, \eta, \delta/2)$

---

*any distribution $\mathcal{D}$ over $\{0,1\}^n$, we define a distribution $B_{\mathcal{D}}$ over $[k] \cup \{0\}$ as: for $0 \leq i \leq k$, $B_{\mathcal{D}}(i) = \mathcal{D}(S_i)$. We call $B_{\mathcal{D}}$ the bucket distribution of $\mathcal{D}$ and $S_i$ the $i^{th}$ bucket.*

Barbarik3 takes as input two distributions $\mathcal{P}$ and $\mathcal{Q}$ defined over the support $\{0,1\}^n$, along with the parameters for closeness($\varepsilon$), farness($\eta$), and confidence($\delta$). On Line 1, Barbarik3 computes the value of $k$ using $\eta$ and the number of dimensions $n$. Then, using DUAL access to $\mathcal{P}$, and SAMP access to $\mathcal{Q}$, Barbarik3 creates bucket distributions $B_{\mathcal{P}}$ and $B_{\mathcal{Q}}$ as in Defn. 1, in the following way: To sample from $B_{\mathcal{P}}$, Barbarik3 first draws a sample $j \sim \mathcal{P}$, then using the DUAL oracle, determines the value of $\mathcal{P}(j)$. Then, if $j$ lies in the $i^{th}$ bucket i.e., $2^{-i} < \mathcal{P}(j) \leq 2^{-i+1}$, the algorithm takes sample $i$ as the sample from $B_{\mathcal{P}}$. Similarly, to draw a sample from $B_{\mathcal{Q}}$, Barbarik3 draws a sample $j \sim \mathcal{Q}$ and then, using the DUAL oracle to find $\mathcal{P}(j)$, finds $i$ such that $j$ lies in the $i^{th}$ bucket, and then uses $i$ as the sample.

Barbarik3 then calls two subroutines, OutBucket (Section 3.4) and InBucket (Section 3.3). The OutBucket subroutine returns an $\theta$-multiplicative estimate of the TV distance between $B_{\mathcal{P}}$ and $B_{\mathcal{Q}}$, the two bucket distributions of $\mathcal{P}$ and $\mathcal{Q}$, with an error of at most $\delta/2$. If it is found on Line 9 that the estimate $\widehat{d}$ is greater than $\varepsilon/2 + \theta$, we know that $d_{TV}(\mathcal{P}, \mathcal{Q}) > \varepsilon/2$ and also that $d_{\infty}(\mathcal{P}, \mathcal{Q}) > \varepsilon$, and hence the algorithm returns Reject. Otherwise, the algorithm calls the InBucket subroutine.

Now suppose that $d_{TV}(\mathcal{P}, \mathcal{Q}) \geq \eta$. Then, for $\varepsilon_2$ (Line 11), it is either the case that $d_{TV}(B_{\mathcal{P}}, B_{\mathcal{Q}}) > \varepsilon_2$ or else $d_{TV}(B_{\mathcal{P}}, B_{\mathcal{Q}}) \leq \varepsilon_2$. In the former case, the algorithm returns Reject on Line 10, and in the latter case the InBucket subroutine returns Reject. In both cases, the failure probability is at most $\delta/2$. Thus Barbarik3 returns Reject on given $\eta$-far input distributions with probability at least $1 - \delta$.

We will now prove the following theorem:

**Theorem 1.** Barbarik3$(\mathcal{P}, \mathcal{Q}, \eta, \varepsilon, \delta)$ *takes in distributions $\mathcal{P}$ and $\mathcal{Q}$ defined over $\{0,1\}^n$, and parameters $\eta \in (0,1]$, $\varepsilon \in [0, \eta/11.6)$ and $\delta \in (0, 1/2]$. Barbarik3 has DUAL access to $\mathcal{P}$, and PCOND+SAMP access to $\mathcal{Q}$. With probability at least $1 - \delta$, Barbarik3 returns*

- Accept *if $d_{\infty}(\mathcal{P}, \mathcal{Q}) \leq \varepsilon$*
- Reject *if $d_{TV}(\mathcal{P}, \mathcal{Q}) > \eta$*

Barbarik3 *has query complexity $\tilde{O}\left(\frac{\sqrt{n}\log(n)}{\eta^3(\eta - 11.6\varepsilon)} + \frac{n}{\eta^2}\right)$, where $\tilde{O}$ hides polylog factors of $\varepsilon, \eta$ and $\delta$.*

## 3.2 Lower bound

The lower bound comes from the paper of Narayanan [30], where it appears in Theorem 1.6. Phrased in the jargon of our paper, the lower bound states that distinguishing between $d_{TV}(\mathcal{P}, \mathcal{Q}) > \eta$ and $d_{\infty}(\mathcal{P}, \mathcal{Q}) = 0$ requires $\tilde{\Omega}(\sqrt{n/\log(n)}/\eta^2)$ samples. Note that the lower bound is shown on a special case ($\varepsilon = 0$) of our problem. Hence the lower bound applies to our problem as well. Furthermore, the lower bound is shown for the case where distribution $\mathcal{P}$ provides full access, i.e., the algorithm can make arbitrary queries to $\mathcal{P}$. This is a stronger access model than DUAL. Since the lower bound is for a stronger access model, it extends to our problem as well.

### 3.3 The InBucket subroutine

In this section, we present the InBucket subroutine, whose behavior is stated in the following lemma.

**Lemma 1.** InBucket$(\mathcal{P}, \mathcal{Q}, k, \varepsilon, \varepsilon_2, \eta, \delta)$ *takes as input two distributions* $\mathcal{P}, \mathcal{Q}$, *an integer* $k$ *and parameters* $\varepsilon, \varepsilon_2, \eta, \delta$. *If* $d_\infty(\mathcal{P}, \mathcal{Q}) \leq \varepsilon$, InBucket *returns* Accept. *If* $d_{TV}(\mathcal{P}, \mathcal{Q}) \geq \eta$ *and* $d_{TV}(B_\mathcal{P}, B_\mathcal{Q}) < \varepsilon_2$, *then* InBucket *returns* Reject. InBucket *errs with probability at most* $\delta$ .

---

**Algorithm 2** InBucket$(\mathcal{P}, \mathcal{Q}, k, \varepsilon, \varepsilon_2, \eta, \delta)$

---

1: $\varepsilon_1 \leftarrow (0.99\eta - 3.25\varepsilon_2 - 2\varepsilon/(1-\varepsilon))/1.05 + 2\varepsilon/(1-\varepsilon)$
2: $m \leftarrow \lceil \sqrt{k}/(0.99\eta - 3.25\varepsilon_2 - \varepsilon_1) \rceil$
3: $\alpha \leftarrow (\varepsilon_1 + 2\varepsilon/(1-\varepsilon))/2$
4: $t \leftarrow \left\lceil \frac{\ln(4/\delta)}{\ln(10/(10-\varepsilon_1+\alpha))} \right\rceil$
5: **for** $t$ iterations **do**
6:      $\Gamma_\mathcal{P} \leftarrow m$ samples from $\mathcal{P}$
7:      $\forall_{i\in[k]}\Gamma_\mathcal{P}^i \leftarrow \Gamma_\mathcal{P} \cap S_i$                            $\triangleright$ $S_i$ is defined in Defn. 1
8:      $\Gamma_\mathcal{Q} \leftarrow m$ samples from $\mathcal{Q}$
9:      $\forall_{i\in[k]}\Gamma_\mathcal{Q}^i \leftarrow \Gamma_\mathcal{Q} \cap S_i$
10:      **for all** $j \in [k]$ s.t. $|\Gamma_\mathcal{P}^j|, |\Gamma_\mathcal{Q}^j| > 0$ **do**
11:          $p \leftarrow \Gamma_\mathcal{P}^j$                                  $\triangleright$ $p$ is an arbitrary sample from the set $\Gamma_\mathcal{P}^j$
12:          $q \leftarrow \Gamma_\mathcal{Q}^j$                                  $\triangleright$ $q$ is an arbitrary sample from the set $\Gamma_\mathcal{Q}^j$
13:          $h \leftarrow \frac{\mathcal{P}(p)}{\mathcal{P}(p)+\mathcal{P}(q)(1+\frac{2\varepsilon}{1-\varepsilon})}$
14:          $\ell \leftarrow \frac{\mathcal{P}(p)}{\mathcal{P}(p)+\mathcal{P}(q)(1+\alpha)}$
15:          $r \leftarrow \left\lceil \frac{2\ln(4mt/\delta)}{(h-\ell)^2} \right\rceil$
16:          $\widehat{c} \leftarrow$ Bias$(Q, p, q, r)$
17:          **if** $\widehat{c} \leq (h+\ell)/2$ **then**
18:              **Return** Reject
19: **Return** Accept

---

**Algorithm 3** Bias$(\mathcal{Q}, p, q, r)$

---

1: **if** $p$ and $q$ are identical **then**
2:      **Return** $0.5$
3: $\Gamma_{\mathcal{Q}_{\{p,q\}}} \leftarrow r$ samples from $\mathcal{Q}_{\{p,q\}}$
4: **Return** # of times $p$ appears in $\Gamma_{\mathcal{Q}_{\{p,q\}}}$

---

InBucket makes extensive use of the PCOND oracle access to $\mathcal{Q}$ via the Bias subroutine, which we describe in the following subsection.

**The** Bias **subroutine**      The Bias subroutine takes in distribution $\mathcal{Q}$, two elements $p, q$ and a positive integer $r$. Then, using the PCOND oracle, Bias draws $r$ samples from the conditional distribution $\mathcal{Q}_{\{p,q\}}$ and returns the number of times it sees $p$ in the $r$ samples. It can be seen that the returned value is an empirical estimate of $\frac{\mathcal{Q}(p)}{\mathcal{Q}(p)+\mathcal{Q}(q)}$. Let the estimate be $\widehat{c_{pq}}$. We use the Hoeffding bound in Prop. 1, and the value of $r$ from Line 15 of Alg. (2) to show that:

$$\Pr\left[\frac{\mathcal{Q}(p)}{\mathcal{Q}(p)+\mathcal{Q}(q)} - \widehat{c_{pq}} \geq \frac{h-\ell}{2}\right] \leq \frac{\delta}{4mt} \qquad \Pr\left[\widehat{c_{pq}} - \frac{\mathcal{Q}(p)}{\mathcal{Q}(p)+\mathcal{Q}(q)} \geq \frac{h-\ell}{2}\right] \leq \frac{\delta}{4mt}$$

Here $t$ represents the number of iterations of the outer loop (Line 4), and $m$ is the number of samples drawn from $B_\mathcal{P}$ and $B_\mathcal{Q}$. Together, there are at most $mt$ pairs of samples that are passed to the Bias oracle. Since in each invocation of Bias, the probability of error is $\delta/4mt$, using the union bound we find that the probability that all $mt$ Bias calls return correctly is at least $1 - \delta/4$ and thus with probability at least $1 - \delta/4$, the empirical estimate $\widehat{c_{pq}}$ is closer than $(h-\ell)/4$ to $\frac{\mathcal{Q}(p)}{\mathcal{Q}(p)+\mathcal{Q}(q)}$.

Henceforth we assume:

$$\left| \widehat{c_{pq}} - \frac{\mathcal{Q}(p)}{\mathcal{Q}(p) + \mathcal{Q}(q)} \right| \leq \frac{h - \ell}{2} \tag{1}$$

### 3.3.1 The Accept case

In this section we will provide an analysis of the case when $d_\infty(\mathcal{P}, \mathcal{Q}) < \varepsilon$. We will now state a proposition required for the remaining proofs, the proof of which we relegate to Appendix A.4.

**Proposition 5.** *Let $\mathcal{P}, \mathcal{Q}$ be distributions and let $p \sim \mathcal{P}$ and $q \sim \mathcal{Q}$. Then,*

1. *If $d_\infty(\mathcal{P}, \mathcal{Q}) < \varepsilon$ then*

$$\frac{\mathcal{Q}(p)}{\mathcal{Q}(p) + \mathcal{Q}(q)} \geq \frac{\mathcal{P}(p)}{\mathcal{P}(p) + (1 + \frac{2\varepsilon}{1-\varepsilon})\mathcal{P}(q)}$$

2. *If $d_{TV}(\mathcal{P}, \mathcal{Q}) > \varepsilon_1$, then for $0 \leq \alpha < \varepsilon_1$, with probability at least $(d_{TV}(\mathcal{P}, \mathcal{Q}) - \alpha)/2$,*

$$\frac{\mathcal{Q}(p)}{\mathcal{Q}(p) + \mathcal{Q}(q)} < \frac{\mathcal{P}(p)}{\mathcal{P}(p) + (1 + \alpha)\mathcal{P}(q)}$$

From our assumption (1), we know that for all invocations of Bias, with probability at least $1 - \delta/4$, $\left| \widehat{c_{pq}} - \frac{\mathcal{Q}(p)}{\mathcal{Q}(p)+\mathcal{Q}(q)} \right| \leq (h - \ell)/2$. Using Prop. 5, and using the value of $h$ given on Line 13, we can see that $\frac{\mathcal{Q}(p)}{\mathcal{Q}(p)+\mathcal{Q}(q)} > h$. From this we can observe that for all invocations of Bias, $\widehat{c_{pq}} > (h + \ell)/2$ and the test does not return Reject in any iteration, hence eventually returning Accept. Thus, in the case that $d_\infty(\mathcal{P}, \mathcal{Q}) < \varepsilon$, the InBucket subroutine returns Accept with probability at least $1 - \delta/4$.

### 3.3.2 The Reject case

In this section we analyse the case when $d_{TV}(\mathcal{P}, \mathcal{Q}) \geq \eta$ and $d_{TV}(B_\mathcal{P}, B_\mathcal{Q}) \leq \varepsilon_2$ and we will show that the algorithm returns Reject with probability at least $1 - \delta$. For the purpose of the proof we will define a set of bad buckets $Bad \subseteq [k]$. Note that bucket $\{0\}$ is not in $Bad$.

**Definition 2.** $Bad = \{i \in [k] : d_{TV}(\mathcal{P}_{S_i}, \mathcal{Q}_{S_i}) > \varepsilon_1 \wedge B_\mathcal{P}(i)/B_\mathcal{Q}(i) \in [5^{-1}, 2]\}$

Suppose we have an indicator variable $X_{r,s}$ constructed as follows: draw $m$ samples from $\mathcal{P}$ and $\mathcal{Q}$, and if the $r^{th}$ sample from $\mathcal{P}$ and the $s^{th}$ sample from $\mathcal{Q}$ both belong to some bucket $b \in Bad$, then $X_{r,s} = 1$ else $X_{r,s} = 0$. Then,

$$\mathbb{E}[X_{r,s}] = \sum_{b \in Bad} B_\mathcal{P}(b) B_\mathcal{Q}(b) > \frac{(B_\mathcal{P}(Bad) + B_\mathcal{Q}(Bad) - 2d_{TV(Bad)}(B_\mathcal{P}, B_\mathcal{Q}))^2}{4K}$$

The inequality is by the application of Prop. 3.

We analyse the expression for the expectation in the following lemma, the proof of which we relegate to Appendix A.2

**Lemma 2.**

$$B_\mathcal{Q}(Bad) + B_\mathcal{P}(Bad) - 2d_{TV(Bad)}(B_\mathcal{Q}, B_\mathcal{P}) > 2\left(0.99\eta - \frac{13}{4}\varepsilon_2 - \varepsilon_1\right)$$

Using Lemma 2 we immediately derive the fact that $\mathbb{E}[X_{r,s}] > \left(0.99\eta - \frac{13}{4}\varepsilon_2 - \varepsilon_1\right)^2/K$. Let $X = \sum_{r,s \in [m]} X_{r,s}$. Given $m$ samples from $\mathcal{P}$ and $\mathcal{Q}$, $\Pr(X \geq 1)$ is the probability that there is at least one bucket in $Bad$ that is sampled at least once each in both sets of samples.

**Lemma 3.** $\Pr(X \geq 1) > 1/5$

The proof can be found in Appendix A.3. $\qquad\square$

Henceforth we will condition on the the event that $X \geq 1$. In such a case, we know that for some $k \in Bad$, there is a sample $p \sim \mathcal{P}_{S_k}$ and a sample $q \sim \mathcal{Q}_{S_k}$. Then for such a pair of samples $(p, q)$, and some $\alpha$, Prop. 5 tells us that with probability at least $(d_{TV}(\mathcal{P}, \mathcal{Q}) - \alpha)/2$ we have

$$\frac{\mathcal{Q}(p)}{\mathcal{Q}(p) + \mathcal{Q}(q)} < \frac{\mathcal{P}(p)}{\mathcal{P}(p) + (1 + \alpha)\mathcal{P}(q)}$$

Using the assumption made in (1), we immediately have that $\widehat{c_{pq}} \leq \frac{\mathcal{Q}(p)}{\mathcal{Q}(p) + \mathcal{Q}(q)} + \frac{h - \ell}{2}$. But from Prop. 5 we have that $\frac{\mathcal{Q}(p)}{\mathcal{Q}(p) + \mathcal{Q}(q)} < \ell$ and hence $\widehat{c_{pq}} < (h + \ell)/2$. Since $d_{TV}(\mathcal{P}, \mathcal{Q}) \geq \varepsilon_1$, we see that if $X \geq 1$, then with probability at least $(\varepsilon_1 - \alpha)/2$, the iteration returns Reject.

Then, using Lemma 3 we see that in every iteration, with probability at least $(\varepsilon_1 - \alpha)/10$, InBucket returns Reject. There are $t$ iterations, where $t$ (line 4) is chosen such that the overall probability of the test returning Reject is at least $1 - \delta/2$.

### 3.4 The OutBucket subroutine

The OutBucket subroutine takes as input two distributions $\mathcal{D}_1, \mathcal{D}_2$ over $k + 1$ elements and two parameters $\theta$ and $\delta$. Then with probability at least $1 - \delta$, InBucket returns a $\theta$-multiplicative estimate for $d_{TV}(\mathcal{D}_1, \mathcal{D}_2)$.

The OutBucket starts by drawing $\max\left(\frac{4(k+1)}{\theta^2}, \frac{8 \ln(4/\delta)}{\theta^2}\right)$ samples from the two distributions $\mathcal{D}_1$ and $\mathcal{D}_2$, and constructs the empirical distributions $\widehat{\mathcal{D}_1}$ and $\widehat{\mathcal{D}_2}$. Then from Prop. 4, we know that with probability at least $1 - \delta$, both $d_{TV}(\mathcal{D}_1, \widehat{\mathcal{D}_1}) \leq \theta/2$ and $d_{TV}(\mathcal{D}_2, \widehat{\mathcal{D}_2}) \leq \theta/2$.

From the triangle inequality we have that,

$$d_{TV}(\widehat{\mathcal{D}_1}, \widehat{\mathcal{D}_2}) \leq d_{TV}(\mathcal{D}_1, \widehat{\mathcal{D}_1}) + d_{TV}(\mathcal{D}_2, \widehat{\mathcal{D}_2}) + d_{TV}(\mathcal{D}_1, \mathcal{D}_2) < \theta + d_{TV}(\mathcal{D}_1, \mathcal{D}_2)$$

and also that,

$$d_{TV}(\mathcal{D}_1, \mathcal{D}_2) \leq d_{TV}(\mathcal{D}_1, \widehat{\mathcal{D}_1}) + d_{TV}(\mathcal{D}_2, \widehat{\mathcal{D}_2}) + d_{TV}(\widehat{\mathcal{D}_1}, \widehat{\mathcal{D}_2}) < \theta + d_{TV}(\widehat{\mathcal{D}_1}, \widehat{\mathcal{D}_2})$$

Thus with probability at least $1 - \delta$, the returned estimate $d_{TV}(\widehat{\mathcal{D}_1}, \widehat{\mathcal{D}_2})$ satisfies $|d_{TV}(\widehat{\mathcal{D}_1}, \widehat{\mathcal{D}_2}) - d_{TV}(\mathcal{D}_1, \mathcal{D}_2)| < \theta$.

**Query and runtime complexity** The number of queries made by OutBucket to $\mathcal{P}$ and $\mathcal{Q}$ is given by $\tilde{O}\left(\frac{n}{\eta^2}\right)$, where $\tilde{O}$ hides polylog factors of $\varepsilon, \eta$ and $\delta$. The number of queries required by InBucket is given by $mtr$. Bounding the terms individually, we see that $m = \tilde{O}\left(\frac{\sqrt{n}}{\eta - 11.6\varepsilon}\right)$, $t = \tilde{O}\left(\frac{1}{\eta}\right)$ and $r = \tilde{O}\left(\frac{\log n}{\eta^2}\right)$. Thus $mtr = \tilde{O}\left(\frac{\sqrt{n} \log n}{(\eta - 11.6\varepsilon)\eta^3}\right)$ and hence the total query complexity is $\tilde{O}\left(\frac{\sqrt{n} \log n}{(\eta - 11.6\varepsilon)\eta^3} + \frac{n}{\eta^2}\right)$.

## 4 Evaluation

To evaluate the performance of Barbarik3 and test the quality of publicly available samplers, we implemented Barbarik3 in Python. Our evaluation took inspiration from the experiments presented in previous work [14, 27], and we used the same framework to evaluate our proposed algorithm. The role of target distribution $\mathcal{P}$ was played by WAPS[4] [23]. WAPS compiles the input Boolean formula into a representation that allows exact sampling and exact probability computation, thereby implementing the SAMP and EVAL oracles needed for our test.

For the role of sampler $\mathcal{Q}(\varphi, \mathbf{w})$, we used the state-of-the-art samplers wSTS and wUnigen3. wUnigen3 [32] is a hashing-based sampler that provides $(\varepsilon, \delta)$ guarantees on the quality of the samples. wSTS [20] is a sampler designed for sampling over challenging domains such as energy barriers and highly asymmetric spaces. wSTS generates samples much faster than wUnigen3, albeit

---

[4]https://github.com/meelgroup/WAPS

without any guarantees on the quality of the samples. To implement PCOND access, we use the Kernel construction from [14]. Kernel takes in $\varphi$ and two assignments $\sigma_1, \sigma_2$, and returns a function $\widehat{\varphi}$ on $m$ variables, such that: (1) $m > n$, (2) $\varphi$ and $\widehat{\varphi}$ are similar in structure, and (3) for $\sigma \in \widehat{\varphi}^{-1}(1)$, it holds that $\sigma_{\downarrow supp(\varphi)} \in \{\sigma_1, \sigma_2\}$. Here $\sigma_{\downarrow supp(\varphi)}$ denotes the projection of $\sigma$ on the variables of $\varphi$.

For the closeness($\varepsilon$), farness($\eta$), and confidence($\delta$) parameters, we choose the values $0.05, 0.9$ and $0.2$. This setting implies that for a given distribution $\mathcal{P}$, and for a given sampler $\mathcal{Q}(\varphi, \mathtt{w})$, Barbarik3 returns (1) Accept if $d_\infty(\mathcal{P}, \mathcal{Q}(\varphi, \mathtt{w})) < 0.05$, and (2) Reject if $d_{TV}(\mathcal{P}, \mathcal{Q}(\varphi, \mathtt{w})) > 0.9$, with probability at least $0.8$. Our empirical evaluation sought to answer the question: How does the performance of Barbarik3 compare with the state-of-the-art tester Barbarik2?

Our experiments were conducted on a high-performance compute cluster with Intel Xeon(R) E5-2690v3@2.60GHz CPU cores. We use a single core with 4GB memory with a timeout of 16 hours for each benchmark. We set a sample limit of $10^8$ samples for our experiments due to our limited computational resources. The complete experimental data along with the running time of instances, is presented in the Appendix B.

### 4.1 Setting A - scalable benchmarks

**Dataset** Our dataset consists of the union of two $n$-dimensional product distributions, for $n \in \{4, 7, 10, \ldots, 118\}$. We have 39 problems in the dataset. We represent the union of two product distributions as the constraint: $\varphi(\sigma) = \bigwedge_{i=1}^{2k}(\sigma_{3k+1} \vee \sigma_i) \wedge \bigwedge_{i=2k+1}^{3k}(\neg\sigma_{3k+1} \vee \sigma_i)$, and the weight function: $\mathtt{w}(\sigma) = \prod_{i=2k+1}^{3k} 3^{\sigma_i}$, where $\sigma_i$ is the value of $\sigma$ at position $i$.

**Results** We observe that in the case of wSTS, Barbarik2 can handle only 12 instances within the sample limit of $10^8$. On the other hand, Barbarik3 can handle all 39 instances using at the most $10^6$ samples. In the case of wUnigen3, Barbarik2 solves 5 instances, and Barbarik3 can handle 17 instances.

Figure 1 shows a cactus plot comparing the sample requirement of Barbarik3 and Barbarik2. The $x$-axis represents the number of benchmarks and $y$-axis represents the number of samples, a point $(x, y)$ implies that the relevant tester took less than $y$ number of samples to distinguish between $d_{TV}(\mathcal{P}, \mathcal{Q}(\varphi, \mathtt{w})) > \eta$ and $d_\infty(\mathcal{P}, \mathcal{Q}(\varphi, \mathtt{w})) < \varepsilon$, for $x$ many benchmarks. We display the set of benchmarks for which at least one of the two tools terminated within the sample limit of $10^8$. We want to highlight that the $y$-axis is in log-scale, thus showing the sample efficiency of Barbarik3 compared to Barbarik2. For every benchmark, we compute the ratio of the number of samples required by Barbarik2 to test a sampler and the number of samples required by Barbarik3. The geometric mean of these ratios indicates the mean speedup. We find that the Barbarik3's speedup on wSTS is $451\times$ and on wUnigen3 is $10\times$.

### 4.2 Setting B - real-life benchmarks

**Dataset** We experiment on 87 constraints drawn from a collection of publicly available benchmarks arising from sampling and counting tasks[5]. We use distributions from the log-linear family. In a log-linear distribution, the probability of an element $\sigma \in \varphi^{-1}(1)$ is given as: $\Pr[\sigma] \propto \exp\left(\sum_{i=1}^n \sigma_i \theta_i\right)$, where $\theta_i \in \mathbb{R}_{>0}^n$. We found that wUnigen3 was not able to sample from most of the benchmarks in the dataset within the given time limit, and hence we present the results only for wSTS.

**Results** We find that Barbarik3 terminated with a result on all 87 instances from the set of real-life benchmarks, while Barbarik2 could only terminate on 16. We present the results of our experiments in Table 1. The first column indicates the benchmark's name, and the second column has the number of dimensions of the space the distribution is defined on. The third and fifth columns indicate the number of samples required by Barbarik2 and Barbarik3. The fourth and sixth columns report the output of Barbarik2 and Barbarik3.

---

[5]https://zenodo.org/record/3793090

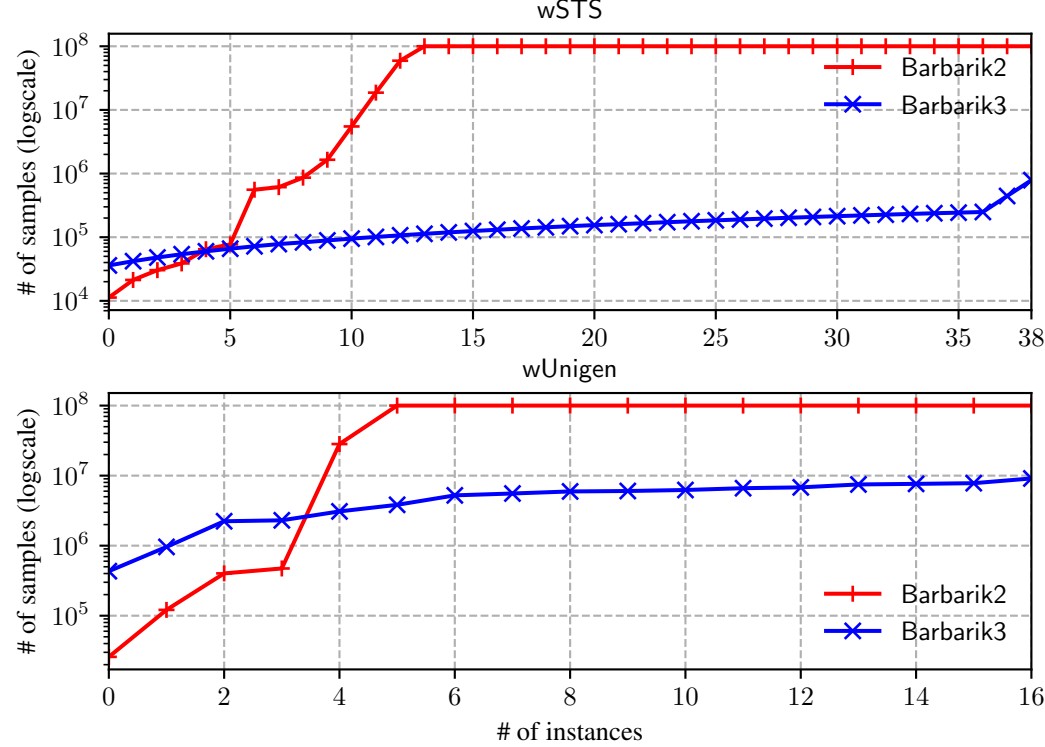

Figure 1: Cactus plot: Barbarik3 vs. Barbarik2. We set the sample limit to be $10^8$, and our dataset consists of 39 benchmarks. The plot shows all the instances where at least one of the two tools terminated within the time limit of 16 hours and sample limit of $10^8$.

Table 1: Runtime performance of Barbarik3. We experiment with 87 benchmarks, and out of the 87 benchmarks we display 15 in the table and we display the full data in Appendix B . In the table 'A' represents Accept, 'R' represents Reject and 'TO' represents that the tester either asked for more than $10^8$ samples or did not terminate in the given time limit of 16 hours.

| | | Barbarik2 | | Barbarik3 | |
|---|---|---|---|---|---|
| Benchmark | Dimensions | Result | # of samples | Result | # of samples |
| SetTest.sk_9_21 | 21 | R | 2817 | R | 58000 |
| Pollard.sk_1_10 | 10 | R | 7606 | R | 36000 |
| s444_3_2 | 24 | R | 848148 | R | 64000 |
| s526a_3_2 | 24 | R | 848148 | R | 64000 |
| s510_15_7 | 25 | R | 12708989 | R | 66000 |
| s27_new_7_4 | 7 | A | 23997012 | R | 30000 |
| s298_15_7 | 17 | R | 38126967 | R | 50000 |
| s420_3_2 | 34 | TO | - | R | 83000 |
| s382_3_2 | 24 | TO | - | R | 64000 |
| s641_3_2 | 54 | TO | - | R | 123000 |
| 111.sk_2_36 | 36 | TO | - | R | 87000 |
| 7.sk_4_50 | 50 | TO | - | R | 115000 |
| 56.sk_6_38 | 38 | TO | - | R | 91000 |
| s820a_15_7 | 23 | TO | - | R | 62000 |
| ProjectService3.sk_12_55 | 55 | TO | - | R | 125000 |

# 5 Conclusion

In this paper, we studied the problem of testing constrained samplers over high-dimensional distributions with $(\varepsilon, \eta, \delta)$ guarantees. For $n$-dimensional distributions, the existing state-of-the-art testing algorithm, Barbarik2, has a worst-case query complexity that is exponential in $n$ and hence is not ideal for use in practice. We provided an exponentially faster algorithm, Barbarik3, that has a query complexity linear in $n$ and hence can easily scale to larger instances. We implemented Barbarik3 and tested the samplers wSTS and wUnigen3 to determine their sample complexity in practice. The results demonstrate that Barbarik3 is significantly more sample efficient than Barbarik2, requiring $450\times$ fewer samples when it tested wSTS and $10\times$ fewer samples when it tested wUnigen3. Since there is a $\sqrt{n}$ gap between the upper bound provided by our work and the lower bound shown in [30], the problem of designing a more sample efficient algorithm or finding a stronger lower bound, remains open.

**Limitations**    For a given farness parameter $\eta$, Barbarik3 requires the value of the closeness parameter $\varepsilon$ to lie in the interval $[0, \eta/11.6)$. In the case of Barbarik2, the previous state-of-the-art test, the permissible values of $\varepsilon$ for a given $\eta$ lie in the interval $[0, \eta/3)$. Thus, Barbarik3 supports testing with only a subset of parameter values that Barbarik2 support.

## Acknowledgments and Disclosure of Funding

We are grateful to the anonymous reviewers of NeurIPS 2022 for their constructive feedback that greatly improved the paper. We thank Ayush Jain and Shyam Narayanan for helpful discussions regarding the paper. We would also like to thank Anna L.D. Latour and Priyanka Golia for their useful comments on the code and the earlier drafts of the paper.

This work was supported in part by National Research Foundation Singapore under its Campus for Research Excellence and Technological Enterprise (CREATE) programme, NRF Fellowship Programme [NRF-NRFFAI1-2019-0004] and Ministry of Education Singapore Tier 2 grant [MOE-T2EP20121-0011], Ministry of Education Singapore Tier 1 Grant [R-252-000-B59-114]. The computational work for this article was performed on resources of the National Supercomputing Centre, Singapore.

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
