# OpenReview forum: "On Scalable Testing of Samplers "
_NeurIPS.cc/2022/Conference — NeurIPS 2022 Accept_

### Official Review · Reviewer_nZVA · 2022-07-10

**Rating:** 6
**Confidence:** 3
**Soundness:** 4 excellent
**Presentation:** 3 good
**Contribution:** 2 fair

**Summary:**

The present work essentially studies tolerant closeness testing of distributions over {1,-1}^n under the assumption that one not only has sample access but also the ability to 1) evaluate the probability mass at any point in the domain ("DUAL access"), and 2) the ability to sample from the distribution conditioned on any subset of size two of the domain ("PCOND access").

Following Meel-Pote-Chakraborty '20, this paper considers a slightly nonstandard notion of tolerant testing where one should whp accept when the two distributions p, q are "multiplicatively close" in the sense that their pdfs are within a factor of $1 \pm \epsilon$ pointwise, and reject when they are $\eta$-far in TV, for $\epsilon$ at most a small multiple of $\eta$.

They give an algorithm with query complexity scaling like $n/\eta^2 + \sqrt{n} / \eta^4$, up to polylog factors. Notably, this avoids the exponential scaling that one would get from traditional closeness testers that only get iid sample access. They also evaluate their algorithm on synthetic and real-world benchmarks and show that their algorithm outperforms a previous baseline, "Barbarik2", which in the worst case has query complexity scaling exponentially in $n$.

**Questions:**

Questions:
- On P. 3 the paper mentions weighted d-DNNFs as yielding one example where DUAL oracle access is possible and says that these are used in the experimental evaluations, but I don't see any subsequent mention of these.
- Relatedly, in the experiments, how did you simulate PCOND and DUAL access?
- Strictly speaking, the guarantees of Barbarik2 and Pacoco are incomparable because the former only assumes conditional sampling access and not DUAL access, right?

Minor comments:
- P. 1 Line 33: "can be take" -> "can take"
- P. 1: While it's formally defined on P. 3, would be helpful to be more specific on P. 1 about what is meant by getting access to the target distribution, because the linear dependence on n is clearly good to be true if one only has sample access.
- P. 4 Line 125: isn't the DUAL oracle being used for both P and Q, not just for P?
- P. 4 Line 133: Line9 -> Line 9
- Would be helpful to state explicitly in Theorem 1 that you are given DUAL and COND access to both distributions.

**Limitations:**

The authors have adequately addressed the limitations, and I don't see any potential negative societal impact from this work.

**Strengths And Weaknesses:**

My main concern is that even with just DUAL and sample access, Canonne and Rubinfeld '14 already gave a simple and optimal tester for this problem. They consider the standard setting of tolerant closeness testing, where we want to distinguish whether $p$ and $q$ are $\epsilon_1$-close in TV or $\epsilon_2$-far in TV. By Corollary 2 in that work (see also Theorem 4.2.10 from Canonne's survey), they achieve the optimal query complexity of $1 / (\epsilon_2 - \epsilon_1)^2$ in this setting,. The point is that it is easy to come up with an unbiased estimator of the TV between p and q using DUAL and sample access (see (2) in that work for the special case of uniformity testing, which can be adapted easily to closeness testing).

In terms of the parameters in this submission, we would take $\epsilon_2 = \eta$ and $\epsilon_1 = \epsilon/2$. Then if $p,q$ are $\epsilon$-multiplicatively-close, then they are $\epsilon_1 = \epsilon/2$-TV-close, in which case the Canonne-Rubinfeld tester would accept, and if $p,q$ are $\eta$-far, then the tester would reject. Notably, their query complexity is $n$-independent, whereas the submission still has some linear dependence on $n$, and furthermore they do not need a PCOND sampler.

This might just be a big misunderstanding on my part, and if so I'd be happy to see clarification on this point in the rebuttal. In terms of positives, I like the practical motivations with which the submission frames the question of tolerant closeness testing, and the proposed algorithm is a clear improvement over Barbarik2 both theoretically and empirically.

---

> ### Author Response · Authors · 2022-08-02
> **Response to Reviewer nZVA**
>
> We thank the reviewer for their detailed review and suggestions.
>
> (Weaknesses and Q3) We do think there has been a misunderstanding: we will highlight the core difference between the access model of Corollary 2 [Canonne and Rubinfeld '14] and our model:
> 	In CR’14, both $P$ and $Q$ offer DUAL access. In contrast, Pacoco has access to distribution $Q$ via the PCOND+SAMP oracles only, while it can access $P$ via the DUAL oracle. We note that Pacoco operates with the same oracle accesses as Barbarik2. The fact that PCOND is a weaker oracle than DUAL contributes to the higher sample requirement.
> To verify the claim that we do not give DUAL access to $Q$, we would like to point out that in the pseudocode,  ${Q}(j)$ (the point probability of element in $j$ in the distribution ${Q}$), is never queried.
> In our revision, we will make this fact clearer and, as suggested, make the oracle accesses explicit in Theorem 1.
>
>
> (Q1, Q2) To simulate DUAL access, we compile the Boolean functions into the weighted dDNNF representation. dDNNFs allow polytime DUAL access. Our implementation follows directly from Barbarik2. In the experimental section, we mention the use of the tool $\mathsf{WAPS}$  for access to $P$. $\mathsf{WAPS}$ is a tool to deal with weighted dDNNFs. We will make this clearer in our revision.
>
> PCOND access is simulated via the use of chain formulas. Given two satisfying assignments to Boolean formula, a chain formula construction allows sampling from the distribution conditioned on the two assignments. Chain formulas preserve the relative probabilities of the two assignments. Our work uses the construction as is from [Meel-Pote-Chakraborty’20]. They provide a detailed discussion on this in Section 2.2 of their paper.  For clarity, we will add the details to the appendix.
>
> (Minor Comment – “P. 4 Line 125: isn't the DUAL oracle being used for both $P$ and $Q$, not just for $P$?”)  On Line 125, we describe the construction of $B_P$ and $B_Q$. To construct $B_P$ and $B_Q$, only $P$ is accessed via the DUAL oracle. Hence we can say that DUAL access is not used on $Q$.

---

> > ### Comment · Reviewer_nZVA · 2022-08-08
> > **Thanks for the clarification!**
> >
> > Thanks for clarifying, it was indeed a misunderstanding on my part! That said, given that we're already assuming DUAL access to one of the distributions, is there a practical reason why one shouldn't assume DUAL access to the other distribution? In any case, I'm happy to raise my score to a 5 for now and possibly a 6 if the authors can provide some more insight into when it's reasonable to assume the different kinds of oracle access.

---

> > > ### Author Response · Authors · 2022-08-09
> > > **Response to additional comments**
> > >
> > >
> > > Sampling from real-life distributions is computationally intractable in general; hence samplers providing guarantees are slow in practice. Guaranteed sampling techniques, such as FPRASes and compilation-based techniques(WAPS), can offer DUAL access; hence, in our experiments, we use WAPS as a DUAL oracle to $P$.
> > >
> > > On the other hand, we are interested in checking whether a given sampler generates distribution close to the target distribution, and as such, we would like to handle a very general class of samplers. Such samplers rely on heuristics to achieve efficiency in practice, such as mutation-based and importance sampling-based methods. For such techniques, it is much easier to engineer the weaker PCOND access than the stronger DUAL, as it would require a way to determine the point probability of a returned sample, which might differ for every technique. In contrast, the PCOND access is sampler agnostic. Indeed, the sampler we test in our experiments, wSTS, cannot provide DUAL access.
> > >
> > > Suppose a user wants to test whether their sampler $G$ really does sample from the target distribution $A$. For the purpose of testing, they would use a guaranteed sampler to sample from $A$ and then use Pacoco to certify if $G$ is close to $A$. The guaranteed sampler cannot be used in practice owing to the much slower sample generation but can be used to test the faster non-guaranteed sampler.
> > >
> > > We hope the above clarifies the reviewer’s concerns and that the reviewer can adjust the score accordingly. We will add discussion to this effect in the final version of the paper.

---

### Official Review · Reviewer_wbEk · 2022-07-10

**Rating:** 6
**Confidence:** 4
**Soundness:** 4 excellent
**Presentation:** 3 good
**Contribution:** 4 excellent

**Summary:**

The paper is concerned with probabilistically validating if samples are close to high-dimensional distributions.  The model, experiments, implementation, and empirical comparisons are all in the model of Chakraborty-Meel (AAAI19,NeurIPS20).  The framework rejects if the TV distance is too large, and accepts if the multiplicative distance is sufficiently small.  This paper focuses on distributions on n-dimensional hypercubes.

It leverages several oracles called COND, PCOND, and DUAL which allow the sampler to access about the true distribution other than point-wise probabilities.  These have been used elsewhere in the community to show improved speed-ups, but the paper does not specifically discuss how one could in practice implement these primitives efficiently, although it seems they have.

For this problem setting, the previous approach required a number of samples exponential the dimension, while the new algorithm requires samples linear in the dimension.  This is accompanied by formal guarantees and proofs.  The empirical results show the implementation's number of samples for a wide array of problems, and the new approach works significantly better.



**Questions:**

N/A

**Limitations:**

No concerns here.

**Strengths And Weaknesses:**

Strengths:
  - formal proof guarantees, with exponential speed-up
  - empirical evidence of speed-up when implemented

Weaknesses:
 -What seems missing however, is a measurement of how well the methods actually perform for a fixed amount of time, or the wall-clock runtime. I understand the number of samples is a key constraint, but the two methods compared may require other auxiliary data structures or apply other oracles which may take more or less time.  Moreover, it could be the theoretical bounds on the previous method are not tight, and it actually performs very accurately with many fewer samples.
  That is, who is testing the testers?
An evaluation of this of some form would have increased my score for the paper.

---

> ### Author Response · Authors · 2022-08-02
> **Response to Reviewer wbEk**
>
>
> We thank the reviewer for their helpful suggestions. In the revision, we will add
> (1) an explanatory note regarding the implementation of the oracles
> (2) details regarding the runtime of Pacoco

---

### Official Review · Reviewer_3NiS · 2022-07-11

**Rating:** 6
**Confidence:** 2
**Soundness:** 3 good
**Presentation:** 2 fair
**Contribution:** 2 fair

**Summary:**

The problem of testing the closeness of distributions asks to distinguish two cases for input distributions $P,Q$: the case that $P,Q$ are $\epsilon$-close in total variation distance and the case that they're $\eta$-far (for $\epsilon \leq \eta$). One setting that has received attention is the setting where the support of the input distributions is (a subset of) the $n$-dimensional hypercube. The most natural setting for a testing algorithm is to only allow black-box sample access to $P$ and $Q$. However, it is known that this requires the sample complexity to be exponential in $n$. In this submission, the authors present an algorithm, Pacoco, that has sample complexity $\tilde{O}(\sqrt{n} \log n / (\eta - 11.6 \epsilon) + n / \eta^2)$. The algorithm uses two types of conditional sampling queries (COND and DUAL): sample access with conditioning on a subset of the support, and querying the probability of elements of the support. The authors compare their algorithm to one of the most recent works, Barbarik2, which also uses conditional sampling (in a possibly weaker model, see questions below) which has exponential sampling complexity. The experiments show for artificially obtained product distributions and many real-world benchmarks that the asymptotic running time outweighs the constants in the running time for $\epsilon = 0.05$, $\eta = 0.9$ and error probability $\delta = 0.2$.

**Questions:**

My rating mainly mirrors that the model assumptions for the various upper and lower bounds from previous work and Pacoco that are discussed in this paper are very unclear to me. Could you elaborate on the exact conditional sampling model / requirements that are assumed by Barbarik2, Pacoco and the lower bounds mentioned? E.g.: Does Barbarik2 assume DUAL oracle access? Is the lower bound derived from [30] assuming DUAL oracle access? More generally, could you group the upper and lower bounds that are mentioned so that all bounds in one group assume exactly the same model?

Why is there a sudden, steep ascent of #samples for Pacoco on wSTS for #instances > 36?

**Limitations:**

-

**Strengths And Weaknesses:**

Strengths:

* The algorithm has nearly linear sampling complexity in $n$. Although the conditional sampling oracle model it uses seems strong, the sampling models were proposed in previous work and are not artificially tailored to the algorithm.
* In summary, the experiments support the claims on the sample complexity and provide some evidence that the constants are not too large.

Weaknesses:

* It seems that the various upper and lower bounds compared in this paper might be somewhat incompatible.
* Only a single set of constants $\epsilon, \eta, \delta$ is chosen for the experiments, and these values are not motivated.
* A running time comparison is lacking.

---

> ### Author Response · Authors · 2022-08-02
> **Response to Reviewer 3NiS**
>
> Response
>
> We thank the reviewer for their detailed review of the experiments and theoretical contribution.
>
>  In the revision, we will add runtime details.
>
> (Q1) Barbarik2 and Pacoco have identical access to the distributions, which is as follows:
> Distribution P can be accessed with the DUAL oracle only.
> Distribution Q can be accessed with the PCOND and SAMP oracles only.
> The lower bound comes from the paper of Narayanan[30], where it appears in Theorem 1.6. Phrased in the jargon of our paper, the lower bound states that distinguishing between $d_{TV}(P,Q) > \eta $ and $d_{\infty}(P,Q) = 0 $ requires $\tilde{\Omega}(\sqrt{n/\log(n)}/\eta^2)$ samples. Note that the lower bound is shown on a special case ($\varepsilon = 0$) of our problem. Hence the lower bound applies to our problem as well.
>
> In [30], the lower bound is shown for the case where distribution P provides full access, i.e., the algorithm can make arbitrary queries to P. This is a stronger access model than DUAL. The lower bound is for a stronger access model, hence it extends to our problem as well.
>
> For clarity, in our revision, we will include a table placing our results among the existing upper and lower bounds.
>
> (Q2) Our test consists of two subtests. The OutBucket test is run first and manages to reject wSTS in most of the cases. In a few instances, OutBucket accepts, and the InBucket test is run. The InBucket test generally requires much more samples. The few instances where InBucket runs contribute to the sharp increase in sample complexity. Here we would like to note that in the plot, the observed sample requirement has been sorted in increasing order along the x-axis, so the few cases where InBucket is called, show up to the right.

---

> > ### Comment · Reviewer_3NiS · 2022-08-05
> > **Reply**
> >
> > This addresses my questions adequately. Thank you! Please take extra care to revise the manuscript and explain the access to the sampling oracles as precisely as in your responses. If other reviewers don't spot a contradiction in the whole picture, I will adjust my score.

---

### Meta-Review · Area_Chair_tYtw · 2022-08-30

**Recommendation:** Accept
**Confidence:** Certain

**Metareview:**

This submission studies (a somewhat non-standard version of) tolerant closeness testing of distributions over the n-dimensional hypercube. Instead of only iid samples, it is assumed that the tester is able to efficiently evaluate the probability mass at any point in the domain and to sample from the distribution conditioned on any subset of size two of the domain. The main result is
an algorithm with query complexity scaling near-linearly in the dimension. Using only iid samples, one would need exponential dependence on dimension. The algorithm is evaluated on synthetic and real-world datasets. It is experimentally shown that their algorithm outperforms a previous baseline, which in the worst case has complexity scaling exponentially in the dimension. Overall, this is an interesting work that appears to meet the bar for acceptance.

**Award:**

No

---

### Decision · Program_Chairs · 2022-09-14

Accept